# Performance of Generative Pretrained Transformer on the National Medical Licensing Examination in Japan

**Yudai Tanaka**[1,2,3☯], **Takuto Nakata**[1,2,3☯], **Ko Aiga**[2☯], **Takahide Etani**[1,4,5],
**Ryota Muramatsu**[1,3], **Shun Katagiri**[1], **Hiroyuki Kawai**[1], **Fumiya Higashino**[1],
**Masahiro Enomoto**[1], **Masao Noda**[6], **Mitsuhiro Kometani**[2], **Masayuki Takamura**[7],
**Takashi Yoneda**[2,8], **Hiroaki Kakizaki**[9], **Akihiro Nomura**[2,7,8,10,11☯]*

**1** School of Medicine, Kanazawa University, Kanazawa, Japan, **2** Department of Health Promotion and Medicine of the Future, Kanazawa University Graduate School of Medicine, Kanazawa, Japan, **3** Department of Molecular and Cellular Pathology, Graduate School of Medical Sciences, Kanazawa University, Kanazawa, Japan, **4** Graduate School of Media and Governance, Keio University, Fujisawa, Japan, **5** Advanced Research Center for Human Sciences, Waseda University, Saitama, Japan, **6** Department of Otolaryngology and Head and Neck Surgery, Jichi Medical University, Tochigi, Japan, **7** Department of Cardiovascular Medicine, Kanazawa University Graduate School of Medical Sciences, Kanazawa, Japan, **8** College of Transdisciplinary Sciences for Innovation, Kanazawa University, Kanazawa Japan, **9** MICIN, Inc., Tokyo, Co, **10** Frontier Institute for Tourism Science, Kanazawa University, Kanazawa, Japan, **11** Department of Biomedical Informatics, CureApp Institute, Karuizawa, Japan

☯ These authors contributed equally to this work.
* anomura@med.kanazawa-u.ac.jp

**Data Availability Statement:** The whole input questions and answers from the model for the 117th NMLE in Japan are listed in the Supplemental Data. The codes used in this study

## Abstract

The remarkable performance of ChatGPT, launched in November 2022, has significantly impacted the field of natural language processing, inspiring the application of large language models as supportive tools in clinical practice and research worldwide. Although GPT-3.5 recently scored high on the United States Medical Licensing Examination, its performance on medical licensing examinations of other nations, especially non-English speaking nations, has not been sufficiently evaluated. This study assessed GPT's performance on the National Medical Licensing Examination (NMLE) in Japan and compared it with the actual minimal passing rate for this exam. In particular, the performances of both the GPT-3.5 and GPT-4 models were considered for the comparative analysis. We initially used the GPT models and several prompts for 290 questions without image data from the 116[th] NMLE (held in February 2022 in Japan) to maximize the performance for delivering correct answers and explanations of the questions. Thereafter, we tested the performance of the best GPT model (GPT-4) with optimized prompts on a dataset of 262 questions without images from the latest 117[th] NMLE (held in February 2023). The best model with the optimized prompts scored 82.7% for the essential questions and 77.2% for the basic and clinical questions, both of which sufficed the minimum passing scoring rates of 80.0% and 74.6%, respectively. After an exploratory analysis of 56 incorrect answers from the model, we identified the three major factors contributing to the generation of the incorrect answers—insufficient medical knowledge, information on Japan-specific medical system and guidelines, and mathematical errors. In conclusion, GPT-4 with our optimized prompts achieved a minimum passing scoring rate in the latest 117[th] NMLE in Japan. Beyond its original design of

are accessible via GitHub (https://github.com/yudaitanaka1026/ChatGPT_NMLE_Japan).

**Funding:** The author(s) received no specific funding for this work.

**Competing interests:** The authors have declared that no competing interests exist.

answering examination questions for humans, these artificial intelligence (AI) models can serve as one of the best "sidekicks" for solving problems and addressing the unmet needs in the medical and healthcare fields.

## Author summary

ChatGPT's remarkable performance has inspired the use of large language models as supportive tools in clinical practice and research. Although it scored well in the US Medical Licensing Examination, its effectiveness in relevant examinations of non-English speaking countries remain unexplored. This study assessed the performance of GPT-3.5 and GPT-4 models in Japan's National Medical Licensing Examination (NMLE). Initially, we used an optimization dataset of 290 questions from the 116th NMLE, and then the GPT-4 model with optimized prompts was tested on 262 questions from the 117th NMLE. The model scored 82.7% for essential and 77.2% for basic and clinical questions, surpassing the minimum passing scoring rates. Incorrect answers were attributed to insufficient medical knowledge, Japan-specific medical system information, and mathematical errors. In conclusion, GPT-4 achieved a minimum passing scoring rate and can be considered a valuable tool for fulfilling the needs of medical and healthcare fields.

## Introduction

In recent decades, artificial intelligence (AI) algorithms have been widely applied in medical and healthcare fields [1]. Currently, the AI algorithms available for clinical applications have been developed using previous rule-based methods as well as recent machine learning (ML) methods including its subfield of deep learning (DL), promoted by the continually increasing availability of computer resources and vast amount of medical data [2]. Consequently, these medical AI products have been implemented to obtain targeted outputs such as the *prediction* of future disease risk, *classification* as diagnostic support, or *generation* of various texts or images using natural language processing (NLP) in medicine [1–3].

NLP is an area of AI that addresses the interaction between human languages and machines [4]. The major roles of NLP in medicine and healthcare include serving as supportive tools in clinical practice and research [3]. Beyond the prediction of certain risk factors or clinical decision-making, NLP assists physicians and researchers to efficiently extract, translate, classify and analyze patients' information and clinical-free text in electronic medical and health records, in addition to dialogue generation and answering medical information [3,4]. The performance of NLP has dramatically improved following the emergence of transformer-based large language models (LLMs). A transformer is a type of neural network model that employs self-attention mechanism, relating multiple positions of a single sequence to compute a representation of the sequence [5]. LLMs are created using advanced ML techniques, especially deep neural networks, trained on enormous amounts of text data from the Internet and other sources [4]. A few notable LLMs include Language Models for Dialog Applications (LaMDA) [6], Pathway Language Model (PaLM) [7], Large Language Model Meta (LLaMA) [8], and Generative Pretrained Transformer (GPT-3) and later models [9–11].

Recently, InstructGPT (GPT-3.5)—a GPT model employing 175 billion parameters with supervised fine-tuning and reinforcement learning from human feedback [10]—and its dialogue-optimized chatbot (ChatGPT) launched in November 2022 have significantly impacted

NLP fields [12]. By predicting the subsequent element of the texts, ChatGPT can comprehend user prompts and generate human-like responses, expressed in ethical, sentimental, logical, and creative manner, without any additional training (*e.g.*, foundation model) [13]. Although GPT is a non-domain-specific LLM, not exclusively intended to be used for medical or health-care fields, recent publications have demonstrated that GPT-3.5 possesses sufficient ability to pass the United States Medical Licensing Examination [14,15]. In contrast, another study reported GPT-3.5's inadequate performance on non-English-based Korean medical questions [16]. Although the performance variation can be attributed to differences in languages, domestic healthcare systems, diagnostic criteria, and treatment strategies, the relationship between these differences and GPT's performance in answering medical questions remains unclear. Furthermore, the performance of the current GPT model (GPT-4) employing an estimated 10 trillion parameters [11] has not yet been fully evaluated on the latest Medical Licensing Examination, which was originally written in non-English texts and held after the completion of GPT-4 model training (August 2022) [17].

Therefore, this study tested the performance of GPT (both GPT-3.5 and GPT-4 models) on the 117[th] National Medical Licensing Examination (NMLE) (held in February 2023 in Japan), which was originally conducted in the Japanese language. In particular, questions from the previous year (116[th] NMLE exam held in February 2022) were used as a model and prompt performance optimization set before using the latest questions (117[th] exam held in February 2023) as a performance testing set to verify whether GPT can qualify for the actual minimal passing rate of this examination. In addition, we analyzed the medical consistency of these model's output results.

## Results

### Improving performance through English translation and optimized prompts in 116[th] NMLE (2022)

Initially, we used the non-image-based questions from 116[th] NMLE in Japan to develop the optimal input prompts for GPT to maximize the scoring rates. We extracted the question data from the 116th NMLE containing 394 questions (originally 400 questions, but six were officially removed from scoring evaluation). Thereafter, we removed questions with image data (n = 104) and analyzed the remaining 290 questions without image data (**Fig 1**).

Using the ChatGPT API powered by GPT-3.5, we initially tested its performance for the original questions in Japanese language. Initially, we obtained the scoring rates of 53.7% (88/164 points) for essential questions and 52.4% (110/210 points) for basic and clinical questions with an output error rate of 5.5%. Accordingly, we used updated prompts to translate the original Japanese NMLE questions into English using GPT-3.5 before inputting them as questions. Although this marginally increased the scoring rates to 60.4% (99/164) for essential questions and 54.8% (115/210) for basic and clinical questions, the output errors increased to 14.8% (**Fig 2**).

To further improve the scoring rates and reduce the errors, we adjusted our prompts for each question type (Basics of Medicine, Clinical Medicine, and Comprehension). In particular, we provided sample outputs and directed the model to translate the questions into plain English and create summaries before answering the questions (**Fig 3**). These optimized prompts improved the scoring rates to 64.6% (106/164) for essential questions and 62.9% (132/210) for basic and clinical questions with a reduced output error rate of 7.6% (22/290). Additionally, we applied the above-optimized prompts to the GPT-4, which demonstrated the scoring rates of 90.9% (149/164) for essential questions and 81.4% (171/210) for basic and clinical questions and a minimal error rate of 1.0% (**Fig 2**). Furthermore, we applied the GPT-4

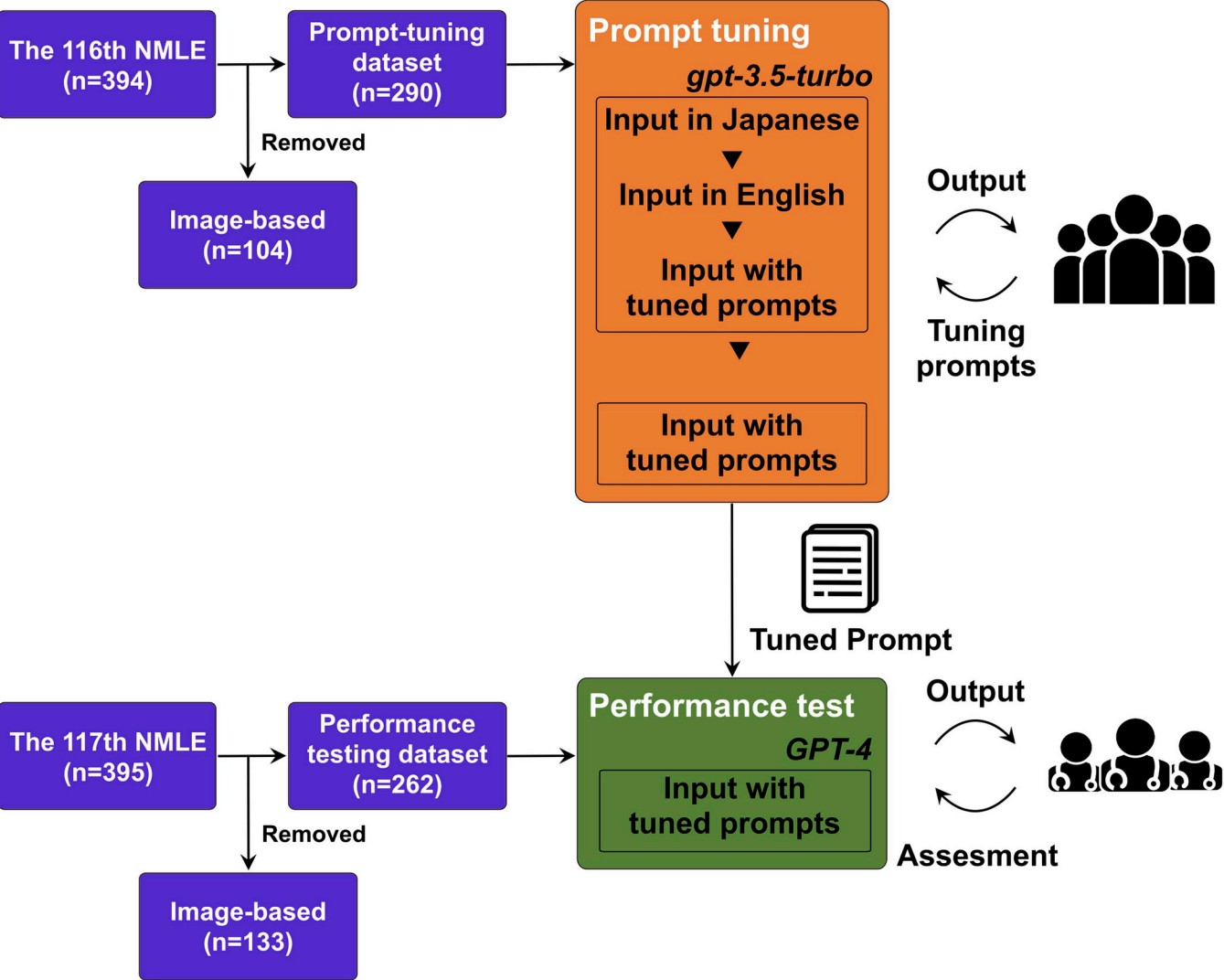

**Fig 1. Study overview.** Questions from the 116[th] NMLE in Japan were used as the prompt optimization dataset and those from 117[th] NMLE were utilized as the performance-testing dataset after removing the image-based questions. During the prompt optimization process, questions from the prompt optimization dataset were input into GPT-3.5-turbo and GPT-4, using simple prompts in both Japanese and English along with optimized prompts in English. Subsequently, we evaluated the outputs from GPT-3.5-turbo and GPT-4 with optimized prompts. After adjusting the prompts, the GPT-4 model with the optimized prompts was tested on the performance-testing dataset (117[th] NMLE).

model with optimized prompts to the entire set of 394 questions (text-only) in the 116[th] NMLE. This optimal model also showed a scoring rate of 88.7% (173/195) for essential questions and 78.5% (233/297) for basic and clinical questions, whose rates were higher than the minimum passing scoring rates as well as those previously reported state-of-the-art model [18].

## GPT-4 performance on 117[th] (2023) NMLE with optimized prompt

Thereafter, we evaluated that the performance of the best model (GPT-4) with optimized prompts for the test set of 262 questions from the 117[th] NMLE in Japan, held in February 4[th] and 5[th], 2023, after the completion of GPT-4 model training in August 2022 (**Fig 1**). With optimized prompts, the best model achieved a scoring rate of 82.7% (129/156) for essential questions and 77.2% (139/180) for basic and clinical questions, both of which qualified the

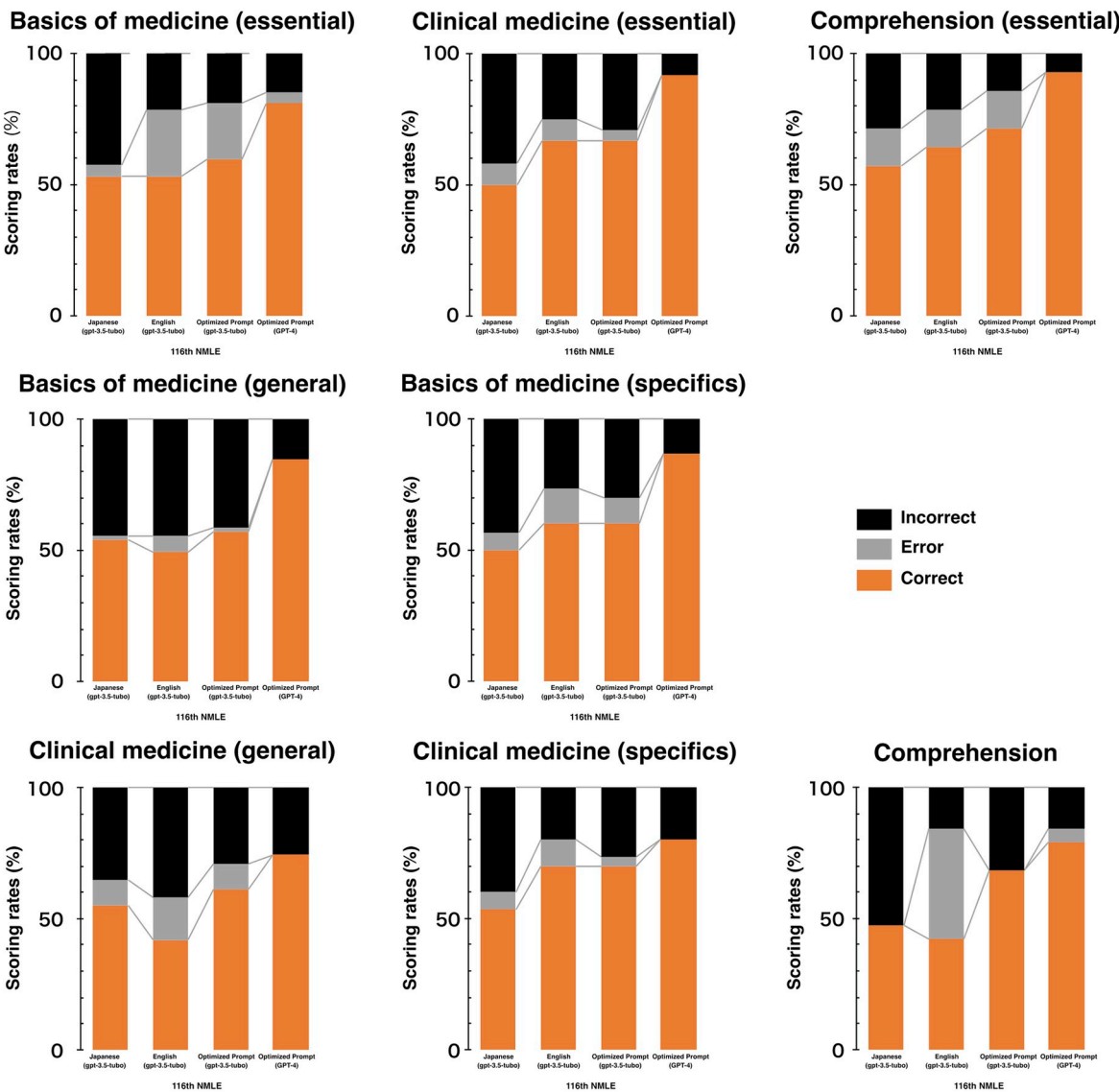

Scoring rates using prompt optimization dataset
(without image data from the 116th NMLE in Japan)

| | | Essential | Basics and Clinical |
|---|---|---|---|
| gpt-3.5-turbo | Japanese | 53.7% | 52.4% |
| | English | 60.4% | 54.8% |
| | Optimized Prompt | 64.6% | 62.9% |
| GPT-4 | Optimized Prompt | 90.9% | 81.4% |

Fig 2. Variations in the scoring rates across languages, prompt adjusting levels, and GPT models. Translating the Japanese questions into English text improved the scoring rates; however, it increased the output error rate. Upon further adjusting the prompts, the scoring rates improved, and the output error decreased. Moreover, switching from the GPT-3.5 model to the GPT-4 model enhanced the scoring rates and almost eliminated errors.

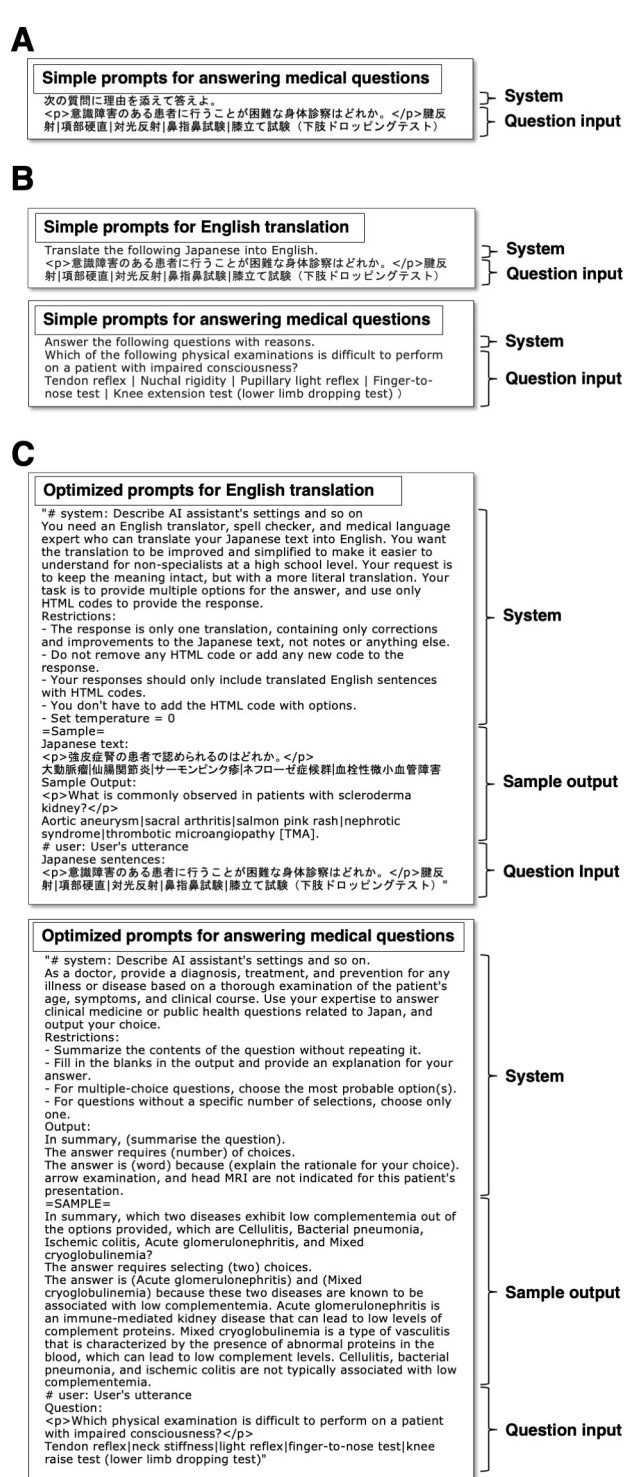

**Fig 3. Examples of prompts for English translation and answering medical questions. A:** A simple "Japanese prompt" used for answering Japanese questions. **B:** Simple "English prompts" used for Japanese-to-English translation and answering translated questions. **C:** Our optimized "English with optimized prompts". The final optimized two-step prompts comprised a "system", "sample output", and "question input" sections. GPT was initially instructed to translate HTML-based Japanese questions into simple, direct, and improved English. In both processes, the system of requirement and an exemplary output scenario were provided within the prompts. In the question input section, the HTML-based Japanese questions were inputted to the English translation process, and sequentially, the English-translated questions were used to obtain the answers of the 117th NMLE questions.

**Table 1. Performance of optimal GPT-4 model with optimized prompt for the 117[th] NMLE in Japan.**

| | Essential | | | Basics and Clinical | | | | |
|---|---|---|---|---|---|---|---|---|
| | Basics of medicine (essential) | Clinical medicine (essential) | Comprehension (essential) | Basics of medicine (general) | Basics of medicine (specifics) | Clinical medicine (general) | Clinical medicine (specifics) | Comprehension |
| No. of questions | 45 | 22 | 15 | 61 | 27 | 36 | 46 | 10 |
| No. of correct answers | 36 | 19 | 12 | 47 | 25 | 22 | 37 | 8 |
| No. of output errors | 1 | 0 | 0 | 0 | 1 | 0 | 0 | 0 |
| No. of incorrect answers | 8 | 3 | 3 | 14 | 1 | 14 | 9 | 2 |
| Correct answer rate | 80.0% | 86.4% | 80.0% | 77.0% | 92.6% | 61.1% | 80.4% | 80.0% |
| Output error rate | 2.2% | 0.0% | 0.0% | 0.0% | 3.7% | 0.0% | 0.0% | 0.0% |
| Score weight | x1 | x3 | | x1 | | | | |
| Total score (scoring rate) | 129/156 (82.7%) | | | 139/180 (77.2%) | | | | |
| Minimum passing scoring rate | 80.0% | | | 74.6% | | | | |

minimum passing scoring rates of 80.0% and 74.6%, respectively (**Fig 2**) [19], and an output error rate of 0.8% (**Table 1**). Notably, we applied the GPT-4 model with optimized prompts to the entire set of 395 questions (text-only) in the 117[th] NMLE, regardless of containing image data (originally 400 questions, but five were officially removed from scoring evaluation). This optimal model attained near-passing levels of 78.5% (157/200) for essential questions and 73.2% (216/295) for basic and clinical questions.

## Exploratory analysis of incorrect GPT-4 responses and their associated explanations

To further enhance the performance of the model, we performed an exploratory analysis of 56 incorrect answers provided by the optimal GPT-4 model with optimized prompts for the 117[th] NMLE questions. As listed in **Table 2**, the three primary factors contributing to the generation of incorrect answers by the model included insufficient medical knowledge (33/56, 58.9%), Japan-specific medical system information (17/56, 30.4%), and mathematical errors (4/56, 7.1%). Concerning the insufficient medical knowledge, the areas of incorrect answers were not specific and spanned across various medical fields. Notably, certain answers were outdated or critically incorrect in current medical contexts (**Fig 4**). In terms of Japan-specific medical system, GPT-4 failed to adequately answer questions related to Japanese medicolegal laws applicable in the medical and healthcare field, guidance from the Ministry of Health, Labor, and Welfare (MHLW) in Japan, and guidelines, especially those related to public health. Additionally, we noted several mathematical errors such as in addition calculations (*e.g.*, the explanation and addition formula were correct, but the answer was wrong) and handling decimal points (because of translation errors from the phrase "rounding to first decimal point" from Japanese).

## Discussion

This study evaluated the performance of GPT on the Japanese Medical Licensing Examination. The results indicate that 1) GPT-4 with optimized prompts cleared the minimal passing rate

**Table 2. Summary of incorrect answers from the optimal model.**

| Total incorrect answer | N = 56 |
|---|---|
| **Insufficient medical knowledge** | **33 (58.9%)** |
| Breast surgery | 1 |
| Dermatology | 2 |
| Emergency medicine | 2 |
| Endocrinology | 6 |
| Gastroenterology | 2 |
| Immunology | 1 |
| Medical interview | 1 |
| Medical procedure | 1 |
| Nephrology | 2 |
| Neurology | 1 |
| Obstetrics and gynecology | 2 |
| Ophthalmology | 1 |
| Pediatrics | 2 |
| Physical examination | 1 |
| Psychiatry | 1 |
| Public health | 1 |
| Rehabilitation | 1 |
| Respiratory medicine | 3 |
| Rheumatology | 1 |
| Urology | 1 |
| **Japan-specific medical system** | **17 (30.4%)** |
| Clinical research | 1 |
| Emergency | 1 |
| Psychiatry | 1 |
| Public health | 14 |
| **Mathematical issues** | **4 (7.1%)** |
| Respiratory | 1 |
| Pediatrics | 1 |
| Cardiology | 1 |
| Medical interview | 1 |
| **Others** | **2 (3.6%)** |
| Issue in English translation | 1 |
| Not providing an answer | 1 |

on the 116[th] (2022) NMLE in Japan; 2) GPT-4 with optimized prompts also qualified the minimum passing scoring rate on the latest 117[th] NMLE (2023); and 3) Inadequate medical knowledge, Japan-specific medical system information, and mathematical errors were the primary factors associated with the incorrect answers generated by the optimal model. Despite the absence of image data in the questions, this study demonstrated the first attempt to use the best available GPT-4 model with optimized prompts to achieve a minimum passing scoring rate for the latest 117[th] NMLE in Japan.

This study provides several conclusions. First, GPT-4 with optimized prompts cleared the minimal passing rate on the 116[th] NMLE in Japan held in February 2022. Although GPT-3.5 achieved a scoring rate of 53.7% for essential questions and 52.4% for basic and clinical questions for Japanese questions, it increased to 60.4% for essential questions and 54.8% for basic and clinical questions after translating the questions into English. As GPT-3, the original GPT-

# A (117B37)

# B (117E44)

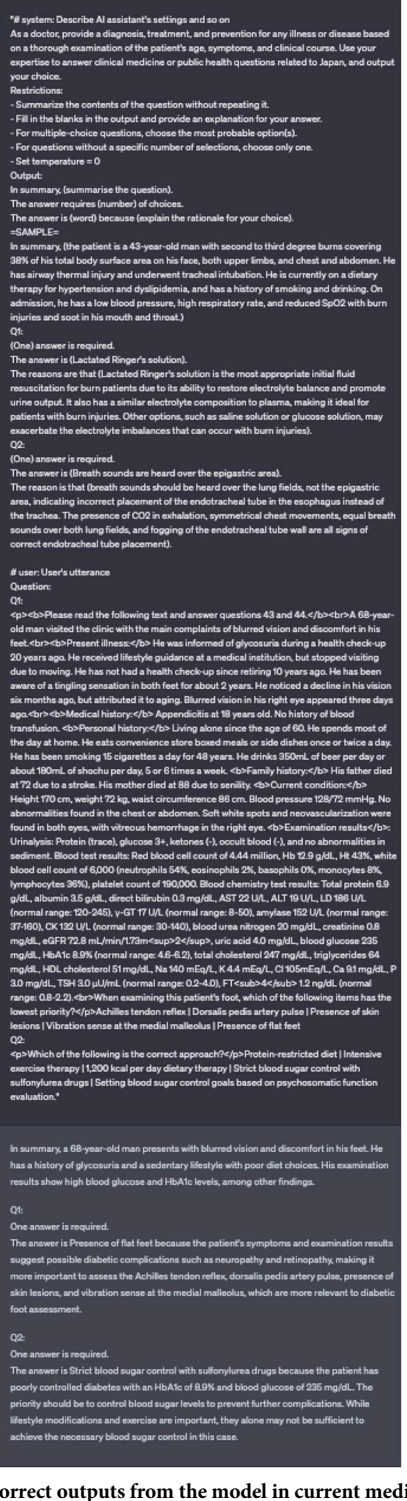

**Fig 4. Examples of potentially outdated or critically incorrect outputs from the model in current medical contexts. A:** A question on the primary treatment for hyperventilation syndrome in the emergency department. The suggestion of paper bag method for raising the carbon-dioxide concentration in the blood has been commonly used in the past, but it is not always the first choice, as it can worsen symptoms in certain patients with secondary hyperventilation, *e.g.*, those with lung diseases causing low blood oxygen levels. In such answers, it seems that

outdated, traditional information can prevail over the latest information, especially if it has been a standard practice over a period and related information is widely available on the Internet. **B:** A question on the initial outpatient treatment for a type-2 diabetes patient with poor control and combined diabetic retinopathy and neuropathy. The long-term treatment goal for diabetes is strict blood sugar control, but in this case, strict blood sugar control with sulfonylurea drugs during the initial treatment may aggravate the risk of diabetic retinopathy, raising strong concerns on GPT-4 answer. (Note: We re-inputted the prompts to obtain the above figures, and the text-contexts of the explanations from GPT-4 differed slightly from that presented in S1 Data, but their meanings are consistent).

3.5, was primarily trained in English, it delivers a higher performance when responding to prompts in English compared to other languages [9]. Similarly, a recent multilingual performance evaluation of GPT-4, an improved version of GPT-3, confirmed that the best performance is more generally obtained with English prompts [11]. In particular, after adjusting our prompts to include a translation procedure into plain English and modifying the output format based on the question type, the scoring rates increased to 64.6% for essential questions and 62.9% for basic and clinical questions. This finding is consistent with previous studies claiming that prompt engineering can improve model task performance [11,20]. These improved correct response rates can be attributed to English being the majority of the language in the training data, *i.e.*, the Internet, used by non-experts [21]. Although the error rate increased to 14.8% upon translating the Japanese questions into English, it notably decreased to 7.6% after adjusting the prompts by including the format of the output. This result suggests that providing samples and standardizing the output format can produce the desired output format and reduce the number of errors. These results are consistent with previous reports [11,20] that prompt engineering can generate more accurate responses to be output in the desired format. Finally, upon applying these optimized prompts to GPT-4, the scoring rates increased to 90.9% for essential questions and 81.4% for basic and clinical questions with higher scoring rates than the previously reported state-of-the-art model prompt, and the error rate plummeted to 1.0%. This significant improvement in performance can be ascribed to the advanced architecture and training of GPT-4 [11].

Second, even in case of the latest 117[th] NMLE (2023), GPT-4 with optimized prompts qualified the actual minimum passing scoring rate. GPT-4 has passed various professional examinations in English, including the practice bar exam with a score in the top 10% of examinees [11]. Although, a previous study reported that GPT-3.5 failed to achieve the minimum passing scoring rates [22], Kasai, et al. first showed the potential performance of the GPT-4 for passing the NLMEs in Japan [22]. Our study, in addition to these previous reports, demonstrated that GPT-4 can also surpass the minimum passing scoring rate of the latest 117th NMLE in Japan using our optimized prompts proposed herein. The current results can be derived from the exquisite combination of essential factors such as English translation and optimally adjusted prompts for obtaining correct answers through the best performance of the latest GPT model.

Third, inadequate medical knowledge, information related to the medical and healthcare system guidelines of Japan, and mathematical errors formed the three major factors of the incorrect answers generated by the best available GPT model with optimized prompts. Among the incorrect answers associated under inadequate medical knowledge, no significant bias was observed for the medical fields relevant to each question. Furthermore, even after providing incorrect answers, the model output plausible medical explanations (so-called hallucinations in LLM outputs [23]). Therefore, even if the model exhibits a performance level that surpasses the minimum score for the NMLE, a broader range of specialized and up-to-date medical knowledge regarding standard treatments should be inputted. In addition, output receivers

should be equipped with professional medical knowledge to assess the correctness of the output. For the Japan-specific system, several incorrect answers were observed, especially in public health-related questions, which are based on Japanese laws, guidelines, and unique systems. Although the GPT-4 delivered improved performance in terms of output differences between the languages, every country should perform their individual localization in terms of the applicable laws and systems considering the language differences. Furthermore, in certain cases related to mathematical issues, the calculation formula in the explanation was correct, but the result and the final answer output were incorrect. Moreover, an instruction of "approximating the decimal place" was not properly comprehended by GPT-4 during the Japanese-to-English translation. As such, calculation problems are reported as one of the areas where LLMs still exhibit relatively low accuracy [24], indicating that calculation problems may be a relatively unsuitable field for the current GPT model.

As discussed, we express strong concerns regarding the use of the current GPT for medical purposes, as OpenAI has already indicated that the models should not be used for providing triage, diagnosis, or treatment options for life-threating issues or severe medical conditions [25]. Indeed, for use in medical settings, an approval must be obtained from regulatory agencies, *e.g.*, a medical device. Moreover, utilizing such technology is already difficult with its several black-box aspects [11]. Various countries have released statements regarding the applications of LLMs in medical fields [26,27]. Although the versatility of these models hinders the verification of their validity and they require enormous computational resources and costs, we believe that the advanced medical foundation AI model [28] can replace task-specific approach AI models and will appear not far off, with scientifically proven clinical efficacy and safety in medical and healthcare fields.

The novelty of this study is that it is the first research to achieve a minimum passing scoring rate using 262 non-image questions in the latest 117th NMLE in Japan with the GPT-4 version with the optimally adjusted prompts. The limitations of this study were as follows. First, we only used questions without image data to evaluate the performance of the best available model with optimized prompts, although it might be fair to assess the ability of the model to pass the examination using all questions, regardless of image data. However, as revealed from the Results, we observed a favorable model performance even upon using the entire question set in the 117th NMLE in Japan. Second, the NMLE in Japan uniquely included strongly not-recommended "contraindication" answer choices within the questions. The MHLW in Japan has set the minimum passing criteria regarding selecting contraindication answer choices to be equal or less than three for the 116th NMLE or two for the 117th NMLE. As the real number of contraindication answer choices were not officially announced by the MHLW, we could not use them in the current performance evaluation. Finally, it should be noted that our optimized prompts were not automatically generated with a specific algorithm. However, our step-wise approach surely improved the performance of answering questions of the NMLE in Japan, that might support the evidence that adapted structured prompts could be one of the options for maximizing scoring performance.

In conclusion, GPT-4 with optimally adjusted prompts achieved a minimum passing scoring rate in the latest 117th NMLE in Japan. In addition, the model scored near-passing levels for the entire test dataset of 395 questions, regardless of medical image data. The upcoming GPT-4 version, which features enhanced image recognition capabilities, will easily qualify the minimum passing scoring rate and achieve top-tier scores, as reported in other English-based examinations [11]. Beyond its original design of answering examination questions for humans, these AI models can be regarded as one of the best "sidekicks" for solving problems and fulfilling the current needs in the medical and healthcare fields.

## Materials and methods

### Study overview

This study evaluated the performance of GPT models on the NMLE in Japan. We utilized both the GPT-3.5 and GPT-4 models (Open AI, Inc., San Francisco, CA, USA). Initially, the questions from the 116[th] NMLE in Japan (February 2022) were used as a model and prompt optimization set to optimize the performance of obtaining the correct answers and explanations. Subsequently, we assessed the performance of the best GPT model (GPT-4) with optimized prompts for answering the questions from the 117[th] NMLE in Japan (February 2023).

### Input source

The questions and answers for the 116[th] NMLE in Japan were obtained from the official website of the MHLW, Japan [29]. For the latest 117[th] NMLE, we manually performed optical character recognition on the original question papers to create input data and extracted the official answers from the MHLW website [19]. The examination comprised six blocks (A–F), with 75 questions in blocks A, C, D, and E, and 50 questions in blocks B and F. Note that six questions in the 116[th] NMLE and five in the 117[th] NMLE were excluded. In addition, all image-containing questions were removed from both the prompt-optimization and the performance-testing datasets, because up till early April 2023, only text-based questions could be used as input to the ChatGPT interface, including the API. The number of image-containing questions was 104 in the 116[th] NMLE and 133 in the 117[th] NMLE. Thereafter, according to the Japanese NMLE scoring method, the remaining questions without image data were classified into the categories of "Essential" and "Basic and Clinical". The 116[th] NMLE in Japan included 47 questions related to basics of medicine (essential), 24 questions of clinical medicine (essential), 14 questions on comprehension (essential), 65 questions regarding basics of medicine (general), 30 questions in basics of medicine (specifics), 31 questions of clinical medicine (general), 60 questions of clinical medicine (specifics), and 19 questions on comprehension. The 117[th] NMLE in Japan comprised 45 questions related to basics of medicine (essential), 22 questions of clinical medicine (essential), 15 questions on comprehension (essential), 61 questions from the basics of medicine (general), 27 questions on the basics of medicine (specifics), 36 of clinical medicine (general), 46 questions related to clinical medicine (specifics), and 10 questions regarding comprehension. Finally, we used 290 questions (without image data) from the 116[th] NMLE and 262 questions (without image data) from the 117[th] NMLE in Japan for analyses. The entire set of 395 text-based questions, irrespective of image data, from the 117[th] NMLE in Japan was considered for the exploratory analysis.

### Generative pretrained transformer

The GPT, developed by OpenAI [13], is a type of AI model used for NLP tasks. Following the research path from the original GPT, GPT-2, and GPT-3, OpenAI's DL approach leverages extensive amounts of data and intensive computation to create increasingly sophisticated and capable language models [17]. ChatGPT has been fine-tuned from the initial GPT-3.5, and later, GPT-4—a LLM trained in early 2022 to produce text [12,30]. GPT-4 is OpenAI's latest and most advanced AI model that can solve difficult problems with greater accuracy [17]. In this study, we used both the GPT-3.5 and the GPT-4 versions.

### Prompt engineering to maximize the scoring rates

We used the 116[th] NMLE in Japan to generate the most suitable prompts for GPT to answer the 117[th] NMLE questions. Using the ChatGPT API, we first instructed GPT to respond to the

original questions in Japanese language. We manually coded the Hyper Text Markup Language (HTML) to represent the bold, italic, superscript, and subscript characters in the original text (**Fig 3A**). Second, we instructed GPT to translate the original Japanese NMLE questions into English using its own capabilities before inputting them as questions (**Fig 3B**). In addition, we compiled and analyzed the output errors. Thereafter, we provided prompts with restriction sentences designed to prevent the reoccurrence of these errors, along with sample outputs illustrating the desired output format. Finally, we inquired GPT to improve the prompt itself. We further refined the prompts using the 116[th] NMLE questions to achieve higher scoring rates and output in the desired format, because prompt fine-adjustment can improve the task accuracy compared to training the entire model [11,20]. The final optimized two-step prompts for the English translation process and the process of answering the medical questions are illustrated in **Fig 3C**, wherein each process comprised "system", "sample output", and "question input" sections. We organized the output examples according to each medical question category (basics of medicine, clinical medicine, and comprehension). In brief, GPT was initially instructed to translate the HTML-based Japanese questions into plain, direct, and improved English, while maintaining the original HTML codes without deleting or adding new text. In both processes, the system of requirement and an exemplary output scenario were provided within the prompts. In the question input section, the HTML-based Japanese questions were inputted for the English translation process, and the English-translated questions were consequently inputted to the process of answering the medical questions (**Fig 3**). To minimize output variability, all input prompts were executed with the temperature parameter set to 0.

These analyses were performed using the ChatGPT API with custom Python code on the Google Colaboratory interface. Specifically, eight investigators (Y. T., T. N., K. A., T. E., R. M., S. K., H. K., and F. H) inputted the questions, choices, and appropriate prompts into GPT and summarized the output answers. We used the GPT-3.5 version GPT3.5-turbo-0301 for the "Japanese", "English", and "English with optimized prompts" analyses, and the GPT-4 model version released on March 14[th], 2023, for the "English with optimized prompts" analysis.

## Outcomes

The target outcome of this study is the scoring rates. We manually compared GPT's output answers with the official answers to determine the correctness of the output answers. Accordingly, the correct answer rate was calculated as the number of correct answers divided by the number of questions. In addition, the scoring rate was evaluated by giving scoring weights to the questions: 1 point for basic medicine questions and 3 points for clinical medicine and comprehension questions, following the scoring criteria of the MHLW, Japan. We defined output errors as a case where a question was not answered, or the number of answers was incorrect. These output errors were handled as wrong answers. To further evaluate performance of the optimized prompt, we compared scoring results obtained with our optimized prompt with those achieved using the state-of-the-art model prompt, as reported in the previous paper [18] during the prompt optimization phase.

## Performance evaluation

In the primary performance evaluation, we assessed the scoring rates for questions without images in the 117[th] NMLE in Japan using the best GPT model (GPT-4) with optimized prompts, which was compared to the actual minimally passing scoring rate on the examination. In the secondary performance evaluation, we examined the scoring rates for all questions in the 117[th] exam using the optimal model and prompts. In addition, the medical

reasonableness of the generated explanations for each answer was assessed by two independent clinical physicians (M.N. and M.K.). Furthermore, we analyzed the content of the incorrect answers along with their explanations to identify the areas in which the application of the current GPT for medicine may be relatively weak.

## Supporting information

**S1 Data. List of questions and corresponding answers for the 117[th] NMLE in Japan, generated by GPT-4 using a tuned prompt.**
(XLSX)

## Acknowledgments

We express our gratitude to Yasuhiro Onogi and Yuichi Miyamae at MICIN, Inc. for their insightful online discussions regarding this project. We thank Dr. Hozumi for dedicating his time to discuss this topic with us. We also thank GPT-4 and Enago English proofreading service for English proofreading.

## Author Contributions

**Conceptualization:** Yudai Tanaka, Takuto Nakata, Ko Aiga, Hiroaki Kakizaki, Akihiro Nomura.

**Data curation:** Yudai Tanaka, Takuto Nakata, Ko Aiga, Takahide Etani, Ryota Muramatsu, Shun Katagiri, Hiroyuki Kawai, Fumiya Higashino, Masahiro Enomoto.

**Formal analysis:** Yudai Tanaka.

**Methodology:** Yudai Tanaka, Takuto Nakata, Hiroaki Kakizaki, Akihiro Nomura.

**Project administration:** Akihiro Nomura.

**Supervision:** Masayuki Takamura, Takashi Yoneda, Hiroaki Kakizaki.

**Validation:** Masao Noda, Mitsuhiro Kometani, Akihiro Nomura.

**Visualization:** Yudai Tanaka, Takuto Nakata, Ko Aiga, Akihiro Nomura.

**Writing – original draft:** Yudai Tanaka, Takuto Nakata, Ko Aiga, Akihiro Nomura.

**Writing – review & editing:** Yudai Tanaka, Takuto Nakata, Ko Aiga, Takahide Etani, Ryota Muramatsu, Shun Katagiri, Hiroyuki Kawai, Fumiya Higashino, Masahiro Enomoto, Masao Noda, Mitsuhiro Kometani, Masayuki Takamura, Takashi Yoneda, Hiroaki Kakizaki, Akihiro Nomura.

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
