## [Decision Letter · Decision Letter 0]

30 Jun 2023

PDIG-D-23-00146

Performance of Generative Pretrained Transformer on the National Medical Licensing Examination in Japan

PLOS Digital Health

Dear Dr. Nomura,

Thank you for submitting your manuscript to PLOS Digital Health. After careful consideration, we feel that it has merit but does not fully meet PLOS Digital Health's publication criteria as it currently stands. Therefore, we invite you to submit a revised version of the manuscript that addresses the points raised during the review process.

Please submit your revised manuscript within 60 days Aug 29 2023 11:59PM. If you will need more time than this to complete your revisions, please reply to this message or contact the journal office at digitalhealth@plos.org. Please include the following items when submitting your revised manuscript:

We look forward to receiving your revised manuscript.

Kind regards,

Ismini Lourentzou

Section Editor

PLOS Digital Health

Journal Requirements:

Additional Editor Comments (if provided):

The paper investigates the performance of GPT-4 on the National Medical Licensing Examination in Japan. It provides a detailed analysis of the predictions and breakdown of results. However, there are concerns regarding differences with contemporary work and the need for clearer explanations and elimination of any inaccuracies in model descriptions and dates. Additionally, more information is needed on the method used for prompt design to enhance the generalizability of the research. Addressing these aspects will strengthen the paper and its suitability for publication.

Reviewers' comments:

Reviewer's Responses to Questions

**Comments to the Author**

1. Does this manuscript meet PLOS Digital Health’s publication criteria? Is the manuscript technically sound, and do the data support the conclusions? The manuscript must describe methodologically and ethically rigorous research with conclusions that are appropriately drawn based on the data presented.

Reviewer #1: Yes

Reviewer #2: No

2. Has the statistical analysis been performed appropriately and rigorously?

Reviewer #1: Yes

Reviewer #2: Yes

3. Have the authors made all data underlying the findings in their manuscript fully available (please refer to the Data Availability Statement at the start of the manuscript PDF file)?

Reviewer #1: Yes

Reviewer #2: Yes

4. Is the manuscript presented in an intelligible fashion and written in standard English?

Reviewer #1: Yes

Reviewer #2: No

5. Review Comments to the Author

Reviewer #1: This paper presents an attempt to use GPT series for solving the national medical licensing examination in Japan. Three prompts were introduced with a detailed analysis of the prediction from GPT-4. 

The overall structure of this paper is great and easy to follow. The breakdown of the prediction gives clear understanding of the GPT-based systems. One of the most serious concerns about this work is the difference between this work and Kasai et al. (2023). I understand that this submission is contemporary to and independent from theirs, but since they published it before your submission, at least the authors need to credit their work and explain the difference between their work and this work. As far as I understand correctly, the finding about the correct answer rates for GPT-3.5 and GPT-4 is pretty similar to their finding (I'm not saying this diminishes the value of this work because this work presents several prompts and reports detailed analysis), so the novelty of this work should be stated more explicitly (compared to theirs).

Reference:

Kasai et al. Evaluating GPT-4 and ChatGPT on Japanese Medical Licensing Examinations. 2023.

https://arxiv.org/abs/2303.18027

Minor comments:

- There were variations of "GPT3.5" and "GPT-3.5".

- Some of the double quotes are "" instead of `` '' (LaTeX). Please check all the occurrences.

- The term "Prompt tuning" has a certain meaning (learning prompt embedding through fine-tuning) in the machine learning literature, so if the authors just intended to use the created dataset for finding a prompt suitable for this task, a different name would better fit.

- I strongly recommend the authors to check the source codes available at GitHub by using a lint tool such as flake8, pep8, etc.

Reviewer #2: I do not think this article includes enough contents to be published as a full journal paper.

There are a number of inaccurate descriptions in this paper:

- GPT-3.5, ChatGPT, and GPT-4 are different models. Each of the language models you used in the experiments should be clearly described. Eg., "GPT-4 powered ChatGPT" (lines 55, 74 and 261) should be "GPT-4, "ChatGPT with GPT-3.5 and GPT-4" should simply be "GPT-3.5 and GPT-4", and so on. There are lots of such mentions.

- Line 96: BERT is a masked language model to obtain token representations in context, which is dis-similar to other generative language models such as LaMDA, PaLM, LLaMA and GPT models.

- Lines 116 and 151: GPT-4 was announced in March 2023, and there is no GPT-4 model trained in August 2022.

- Line 135: What does "output error" mean? There is no explanation about the difference between incorrect answers and output errors. Worth than that, line 351 says "We define the output errors as incorrect answers", but they should be different as Figure 2 shows them differently.

There are a number of papers that investigated the ability of ChatGPT and GPT-4 by applying them to various problems. This paper is one of such an attempt. The originl part of this paper is the method for designing the prompts to tune the LLMs to solve the Japanese medical exams. While examples are shown in appendix, no detail of the ways to obtain the optimal prompts is described. It is good if this method can be generalized. However, if this is done in a heuristical way by hand, the readers cannot learn from this research.

6. PLOS authors have the option to publish the peer review history of their article (what does this mean?). If published, this will include your full peer review and any attached files.

**Do you want your identity to be public for this peer review?** For information about this choice, including consent withdrawal, please see our Privacy Policy.

Reviewer #1: No

Reviewer #2: No

---

## [Decision Letter · Decision Letter 1]

17 Oct 2023

PDIG-D-23-00146R1

Performance of Generative Pretrained Transformer on the National Medical Licensing Examination in Japan

PLOS Digital Health

Dear Dr. Nomura,

Thank you for submitting your manuscript to PLOS Digital Health. After careful consideration, we feel that it has merit but does not fully meet PLOS Digital Health's publication criteria as it currently stands. Therefore, we invite you to submit a revised version of the manuscript that addresses the points raised during the review process.

Please submit your revised manuscript within 30 days Nov 16 2023 11:59PM. If you will need more time than this to complete your revisions, please reply to this message or contact the journal office at digitalhealth@plos.org. Please include the following items when submitting your revised manuscript:

We look forward to receiving your revised manuscript.

Kind regards,

Imon Banerjee

Section Editor

PLOS Digital Health

Journal Requirements:

2. Please send a completed 'Competing Interests' statement, including any COIs declared by your co-authors. If you have no competing interests to declare, please state "The authors have declared that no competing interests exist". Otherwise please declare all competing interests beginning with twhe statement "I have read the journal's policy and the authors of this manuscript have the following competing interests:"

4. We ask that a manuscript source file is provided at Revision. Please upload your manuscript file as a .doc, .docx, .rtf or .tex.

5. Please provide separate figure files in .tif or .eps format only and remove any figures embedded in your manuscript file. Please also ensure that all files are under our size limit of 10MB.

Additional Editor Comments (if provided):

Please update the source code in Github to help the community.

Reviewers' comments:

Reviewer's Responses to Questions

**Comments to the Author**

1. If the authors have adequately addressed your comments raised in a previous round of review and you feel that this manuscript is now acceptable for publication, you may indicate that here to bypass the “Comments to the Author” section, enter your conflict of interest statement in the “Confidential to Editor” section, and submit your "Accept" recommendation.

Reviewer #1: All comments have been addressed

2. Does this manuscript meet PLOS Digital Health’s publication criteria? Is the manuscript technically sound, and do the data support the conclusions? The manuscript must describe methodologically and ethically rigorous research with conclusions that are appropriately drawn based on the data presented.

Reviewer #1: Partly

3. Has the statistical analysis been performed appropriately and rigorously?

Reviewer #1: N/A

4. Have the authors made all data underlying the findings in their manuscript fully available (please refer to the Data Availability Statement at the start of the manuscript PDF file)?

Reviewer #1: Yes

5. Is the manuscript presented in an intelligible fashion and written in standard English?

Reviewer #1: Yes

6. Review Comments to the Author

Reviewer #1: Thank you for updating the manuscript. Almost all the concerns I raised were correctly addressed.

In the response letter, you wrote "However, since they have not shown any detailed prompts they used for the analyses in the preprint manuscript (even for their version 2 paper), we could not replicate or confirmed validity of any of their results." but you can find the prompts at their GitHub repository (https://github.com/jungokasai/IgakuQA/blob/main/scripts/prompts/prompt.jsonl), which was clearly mentioned from their paper. 

Also, I recommended you to check your Python code at GitHub, and you wrote "Thank you for your advice. We again arranged the source codes at GitHub.", but the source code was not updated at all, so I withhold recommending the paper as a full acceptance (you wrote you arranged your code on GitHub, but it is not true. Maybe you forgot to push?).

https://github.com/yudaitanaka1026/ChatGPT_NMLE_Japan

7. PLOS authors have the option to publish the peer review history of their article (what does this mean?). If published, this will include your full peer review and any attached files.

**Do you want your identity to be public for this peer review?** For information about this choice, including consent withdrawal, please see our Privacy Policy. 

Reviewer #1: No

---

## [Decision Letter · Decision Letter 2]

6 Dec 2023

PDIG-D-23-00146R2

Performance of Generative Pretrained Transformer on the National Medical Licensing Examination in Japan

PLOS Digital Health

Dear Dr. Nomura,

Thank you for submitting your manuscript to PLOS Digital Health. After careful consideration, we feel that it has merit but does not fully meet PLOS Digital Health's publication criteria as it currently stands. Therefore, we invite you to submit a revised version of the manuscript that addresses the points raised during the review process.

Please submit your revised manuscript within 60 days Feb 04 2024 11:59PM. If you will need more time than this to complete your revisions, please reply to this message or contact the journal office at digitalhealth@plos.org. Please include the following items when submitting your revised manuscript:

We look forward to receiving your revised manuscript.

Kind regards,

Imon Banerjee

Section Editor

PLOS Digital Health

Journal Requirements:

1. We have noticed that you have uploaded Supporting Information files, but you have not included a list of legends. Please add a full list of legends for your Supporting Information files after the references list.

Additional Editor Comments (if provided):

Please update the source code in the GitHub based on the review suggestion. Also compare the SoA model and present the results.

Reviewers' comments:

Reviewer's Responses to Questions

**Comments to the Author**

1. If the authors have adequately addressed your comments raised in a previous round of review and you feel that this manuscript is now acceptable for publication, you may indicate that here to bypass the “Comments to the Author” section, enter your conflict of interest statement in the “Confidential to Editor” section, and submit your "Accept" recommendation.

Reviewer #1: (No Response)

2. Does this manuscript meet PLOS Digital Health’s publication criteria? Is the manuscript technically sound, and do the data support the conclusions? The manuscript must describe methodologically and ethically rigorous research with conclusions that are appropriately drawn based on the data presented.

Reviewer #1: Partly

3. Has the statistical analysis been performed appropriately and rigorously?

Reviewer #1: No

4. Have the authors made all data underlying the findings in their manuscript fully available (please refer to the Data Availability Statement at the start of the manuscript PDF file)?

Reviewer #1: Yes

5. Is the manuscript presented in an intelligible fashion and written in standard English?

Reviewer #1: Yes

6. Review Comments to the Author

Reviewer #1: Thank you for taking the time to update your manuscript.

However, I can hardly find where you treat the Kasai et al.'s work adequately. If you think the optimization of the prompts is the contribution, you need to compare your prompt against theirs, but Table 1 contains only the results with the optimized prompt (I cannot find the baseline results for example with Kasai et al.'s prompt, nor a baseline result without your optimization). 

Also, I thank you to check that the GitHub repository is correctly updated, but I can find that the update is only for fixing typos. I specifically asked you to use some style-checking tools like flake8, and you said you did so, but there seems no indication that you did so. The source code seems like a script that can be run with Google Colab, but I strongly recommend you to use some lint tools to check the source code.

7. PLOS authors have the option to publish the peer review history of their article (what does this mean?). If published, this will include your full peer review and any attached files.

**Do you want your identity to be public for this peer review?** For information about this choice, including consent withdrawal, please see our Privacy Policy. 

Reviewer #1: No

---

## [Editor Report · Decision Letter 3]

19 Dec 2023

Performance of Generative Pretrained Transformer on the National Medical Licensing Examination in Japan

PDIG-D-23-00146R3

Dear Dr. Nomura,

We are pleased to inform you that your manuscript 'Performance of Generative Pretrained Transformer on the National Medical Licensing Examination in Japan' has been provisionally accepted for publication in PLOS Digital Health.

Best regards,

Imon Banerjee

Section Editor

PLOS Digital Health

I like to congratulate the authors to revise the paper based on suggested comments.